# Why Consider Geomorphology in River Rehabilitation?

Hervé Piégay [1,*], Fanny Arnaud [1], Barbara Belletti [1], Mathieu Cassel [1], Baptiste Marteau [1], Jérémie Riquier [2], Christophe Rousson [1] and Daniel Vazquez-Tarrio [3]

1   EVS (Environnement-Ville-Société), ENS Lyon, CNRS UMR 5600, F-69362 Lyon, France; barbara.belletti@ens-lyon.fr (B.B.)
2   EVS (Environnement-Ville-Société), University Jean Monnet Saint-Étienne, CNRS UMR 5600, F-42023 Saint-Étienne, France
3   Department of Geodynamics, Stratigraphy and Paleontology, Geology Faculty, Complutense University of Madrid, 28040 Madrid, Spain
*   Correspondence: herve.piegay@ens-lyon.fr

**Abstract:** River rehabilitation and ecological engineering are becoming critical issues for improving river status when ecological habitats and connectivity have been altered by human pressures. Amongst the range of existing rehabilitation options, some specifically focus on rebuilding fluvial forms and improving physical processes. The aim of this contribution is to illustrate how geomorphological expertise and process-based thinking contribute to river rehabilitation success. This semantic contribution is intended to feed the rehabilitation debate, particularly concerning the design of actions and the proposed references for monitoring target reaches and evaluating rehabilitation effects empirically. This article is also based on lessons learned from practical cases, mainly in gravel-bed rivers. Geomorphic understanding is needed at a local level to achieve an adequate diagnosis of river functioning, estimate human impacts and potential remnant river responsiveness, and to assess the gains and risks from rehabilitation, as well as to appraise success or failure through several pre- and post-project assessment strategies. Geomorphological studies can also be upscaled in a top-down manner (from high-order controls to small-scale processes, understanding detailed processes in their regional or basin-wide context), providing large-scale information at the regional, national, or even global level, information that can be used to diagnose the health of riverscapes in relation to local site-specific contexts. As such, geomorphological studies support strategic planning and prioritization of rehabilitation works according to specific contexts and river responsiveness, so as to move from opportunistic to objective-driven strategies.

**Keywords:** river restoration; river improvement; diagnosis; evaluation; monitoring; upscaled approach; process-based understanding

## 1. Introduction

Given the widespread alterations to natural systems over recent centuries, conservation and prevention measures alone will not be enough to support ecosystem functions and services [1]. Our dependence on healthy ecosystems has made river rehabilitation a critical strategy for improving river status when ecological habitats and connectivity have been altered by human pressures. However, attempts aiming to implement river rehabilitation through discipline-bound 'hard' engineering applications have failed to fulfil their ambitions [2]. Conversely, a growing number of cross-disciplinary examples have demonstrated the benefits of integrative river science [3,4] and process-based understanding for assuring the positive effects of rehabilitation measures over the long term.

When river rehabilitation first emerged as a question of interest for ecologists, the expected response was primarily biological. However, the topic is currently becoming more and more a social question, because rehabilitation is expected to improve ecosystem services, goods, and well-being, for both humans and non-humans [5,6], such as, e.g., nutrient

retention, water quality improvement [7], protection of infrastructures [8], spawning sites for high-value recreational fish species [9], or flood mitigation [10]. Improving such services can rely on different strategies, among which those acting on channel morphology and hydrology stand out as being particularly meaningful, once good water quality is assured.

The morphological measures intended for the improvement of river status are fairly diverse. We can redesign fluvial forms or their characters to increase habitat diversity, from the very local scale of a single geomorphic feature (typically: riffle, pool, bar) up to the entire floodplain (e.g., modification of substrate conditions or channel geometry along a river reach; creation of lower and more frequently flooded alluvial margins; setting-back of levees; recovering of more dynamic and complex channel patterns such as meanders or multiple threads; reconnection of former side channels). We can also act on controlling factors such as sediment supply through bank re-erosion, dam/weir removal, gravel augmentation below dams in sediment-starved gravel-bed rivers, or improvement of the flow regime through the design of eco-morphogenic flushing flows [3].

We expect that morphological measures will improve river corridor habitats, and then consequently ecological communities. Although this assumption is strong, it needs a careful appraisal to be verified, or even generalized, because of non-cyclic patterns of biophysical processes and responses. The effects of pressures are not symmetrically balanced by the effects of rehabilitation, and special attention must be given to the fact that morphological measures do not systematically imply a positive ecological response. Morphological measures are also critical when considering the sustainability of rehabilitation because they act on river energy, resistance, and responsiveness, which have direct implications on river adjustments and future channel states. Many examples of unsuccessful rehabilitation actions have been reported by scientists (e.g., [11–13]). Most of the time, such failures are related to a lack of a priori knowledge of the morphodynamics and geomorphological context, and the actions are therefore not adapted to the situation, and the expected recovered habitats are rapidly lost or never appear. There is also often a lack of well-defined objectives for guiding rehabilitation measures and assessing their potential success [14–17].

All rivers are not equally sensitive or responsive to change, so all rehabilitation measures are not appropriate for all rivers. When rivers are highly responsive, rehabilitation should be mainly focused on processes, and promoting self-rehabilitation is usually an adequate strategy. When rivers are not very reactive over a certain period of time (e.g., a few decades if we refer to measures that are supposed to be sustainable), acting on their form can be recommended. Hence passive and active physical rehabilitations can both be advocated, but each one within its specific and more suitable contexts.

The aim of this article is to illustrate, from literature review and lessons learned from case studies conducted by the research team mainly in gravel-bed rivers, how morphological approaches can contribute to river rehabilitation success. Morphological understanding can feed the debate at a theoretical level when considering the definition of actions (Section 2), at a local project level during the diagnosis (Section 3), risk assessment (Section 4), and success evaluation (Section 5) steps, and at an up-scaled level (Section 6) to provide the information required to diagnose river health conditions and assess potential responses to river rehabilitation and subsequent failure or success. Different issues are considered according to the rehabilitation phase (Figure 1).

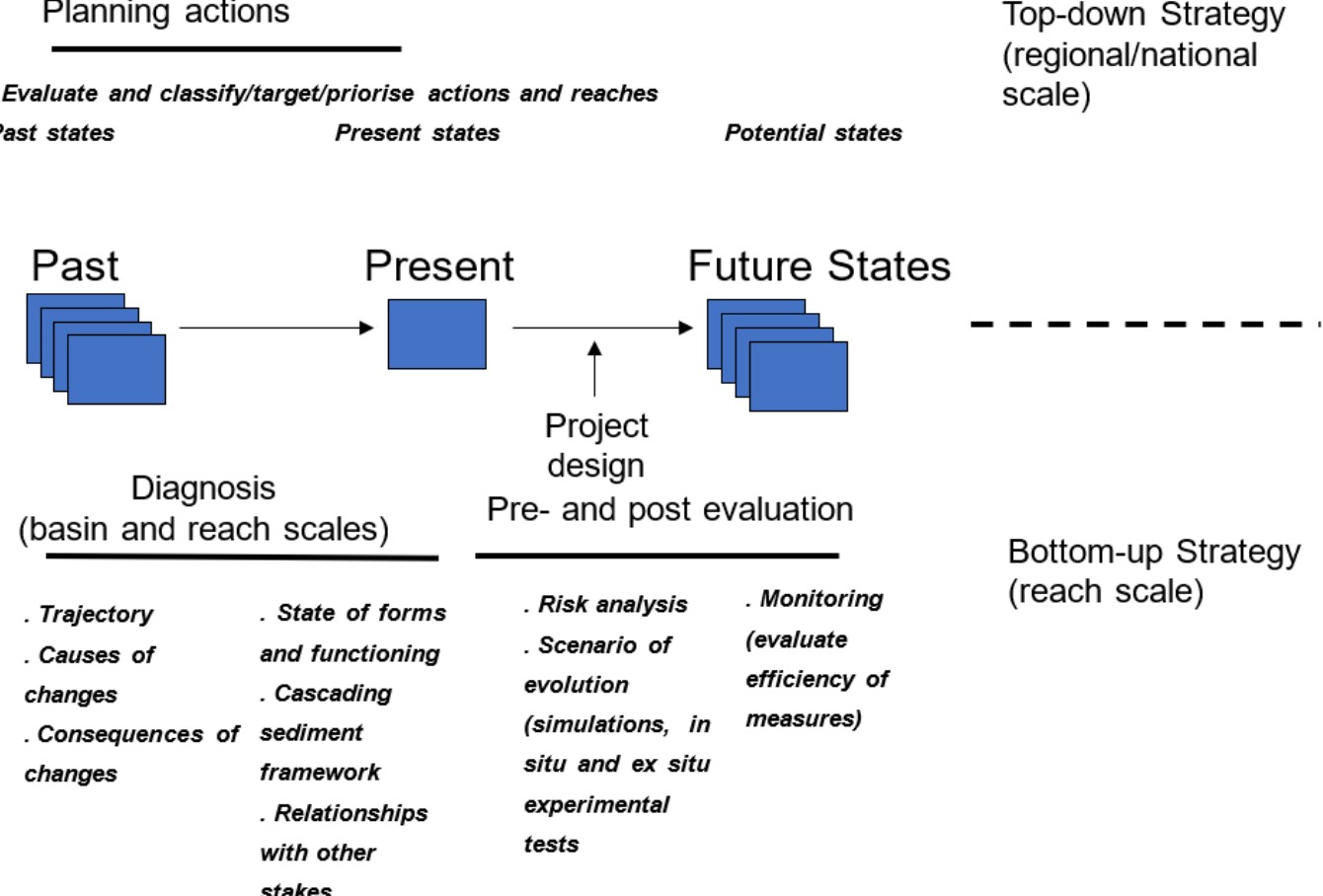

**Figure 1.** Different geomorphic knowledge needs when rehabilitating rivers: diagnosis to understand the alterations and orders of magnitude of active processes (Section 3), project design for evaluating risks (Section 4) or success (Section 5), or even top-down approaches for classifying, targeting, and prioritising actions at a large scale (Section 6) (from Piégay et al. 2016 [18]; reproduced by permission of Elsevier, 2023).

## 2. Reflecting on Geomorphology to Improve Rivers: The Emerging Field of River Repair

Considering rivers through a geomorphologist's perspective has contributed to the debate on "*what is river restoration*". It has shed light on the utopian aim of recovering a pre-disturbance structural and functional state, because rivers are reactive systems responding to a complex set of disturbances within an even more complex space–time framework [19,20]. The return to a pre-disturbance state is now recognized as impractical because of: (i) the irreversibility of many anthropogenic alterations to aquatic ecosystems [21], (ii) the need to reconcile human and environmental needs [22], and (iii) the impossible task of defining a "pre-disturbance" or "pristine" state [23]. Indeed, river channel morphology observations made before the major regulation phases, notably in Europe, were already altered by a longstanding history of deforestation and pastoralism dating back to the Neolithic period [24]. Restoration appears as a dream that is almost impossible to achieve, whereas rehabilitation, in the sense given by [25], is a target that is reachable [26]. Rehabilitation is about repairing damaged ecosystem functions, improving river status, and working towards an alternative steady-state [27], but also about setting objectives that are biophysically possible and socially acceptable [2]. Surprisingly, and despite scientists having proposed some clear definitions of *restoration* and *rehabilitation* in the 1990s (see Table I in [23]), most of the initiatives in the world aiming to improve or repair rivers are still labelled as restoration, despite fitting the definition of rehabilitation ([27]). Moreover,

some of the terms describing river improvement do not always match that classification. For instance, environmental flows used to attenuate the impact of dams are considered as *mitigation* measures, whereas gravel augmentation downstream of dams is often labelled as *restoration*. Sustainable management of dams can be roughly described as a long-term mitigation measure if it is repeated through time, but alternative solutions could be chosen that would better fit the purpose of rehabilitation. For instance, dam flushing practices can be modified to minimise their environmental effects, or to help with vegetation maintenance in non-adjusted reaches within which vegetation encroachment is not permitted because of flood risk. This illustrates the complex story of terminology, and its dissemination, appropriation, and evolution.

Doing things better (i.e., improving the physical quality of a river in a more pragmatic manner) relies on the recognition that rivers are adjusting systems that follow trajectories; future conditions may not resemble past conditions, particularly within the short-term context of the Anthropocene. River systems can react to complex sets of drivers—so-called press disturbances—while responding to pulses of perturbations corresponding to critical flood events [28]. These characteristics highlight the numerous properties of rivers, such as resistance or resilience, sensitivity or responsiveness, and vulnerability. On the basis of a system's responsiveness, it is possible to define whether to act on forms or on processes in order to improve a river's status, to consider process-based measures aimed at increasing geomorphic complexity, and to identify sustainable measures that would be potentially successful in terms of river improvement.

Living with rivers (rather than fighting them) by promoting nature-based solutions forms the basis of geomorphic process-based rehabilitation and mitigation. This concept gave rise to the development of what [2] called the 'emerging process of river repair', and what today lays the foundation for the emerging technical discipline of riverine science and engineering [29].

## 3. Understanding Alterations and Orders of Magnitude of Active Processes

Morphological alterations to rivers must be assessed during the diagnosis phase, to assess their magnitude, their consequences on habitats, and how fast they are likely to occur. These assessments are needed to understand potential ecological damages and anticipate channel responsiveness to rehabilitation actions.

The assessment of past channel changes is a way to better understand controlling factors in space and time, which then allows us to consider what the potential solutions to improve present conditions might be [30–32]. There is a link between upstream changes and potentially cascading effects of changes downstream. In the case of a change in sediment delivery, e.g., by damming, gravel mining activity, or land use change in the catchment, a time lag of several years or decades can be observed between the drivers and the downstream effects [33,34]. In the case of peak flow alteration, a downstream channel response may occur all along the reach as long as hydrological conditions are not significantly changed by the contribution of a tributary. However, the response is not systematically linked to changes in the sediment or flood regime. Drivers can also be local without any upstream changes (e.g., channel straightening or deepening), with the adjusting variables all being interdependent. Another classical example corresponds to the case of in-channel vegetation encroachment, which is an adjustment property linked to ecological conditions controlling recruitment, growth, and resistance, and may be associated with climatic changes or flow seasonality, independent of any change in flood or sediment regime.

Different strategies may be used to identify drivers of change. Comparative approaches applied within an appropriate space–time framework can be powerful for testing and ranking causal factors. They can be based on comparing a set of basins or river reaches, or on exploring the longitudinal pattern on a given river reach to detect change in downstream trends [35]. Such a process-based hierarchical diagnosis was conducted on the Lower Ain River (France) downstream from the Allement dam. Rollet et al. [36]

assessed the impacts of sediment starvation on channel shifting, bed incision, and bed coarsening, and the ecological consequences on riparian forest, floodplain lake vegetation, and fish communities. The causal diagnosis was spatially and temporally explicit in that it identified the dam effects in the longitudinal dimension amongst the other drivers affecting the whole reach (e.g., grazing decline and channel regulation). This multicriteria diagnosis (Figure 2) provided a consistent basis for river managers to implement both curative and preventive actions: the former channel rehabilitation measures were combined with gravel augmentation in the upstream reach that was already affected by sediment starvation, while the free-meandering reaches in the downstream section were preserved within an erodible corridor avoiding any new bank protection [37].

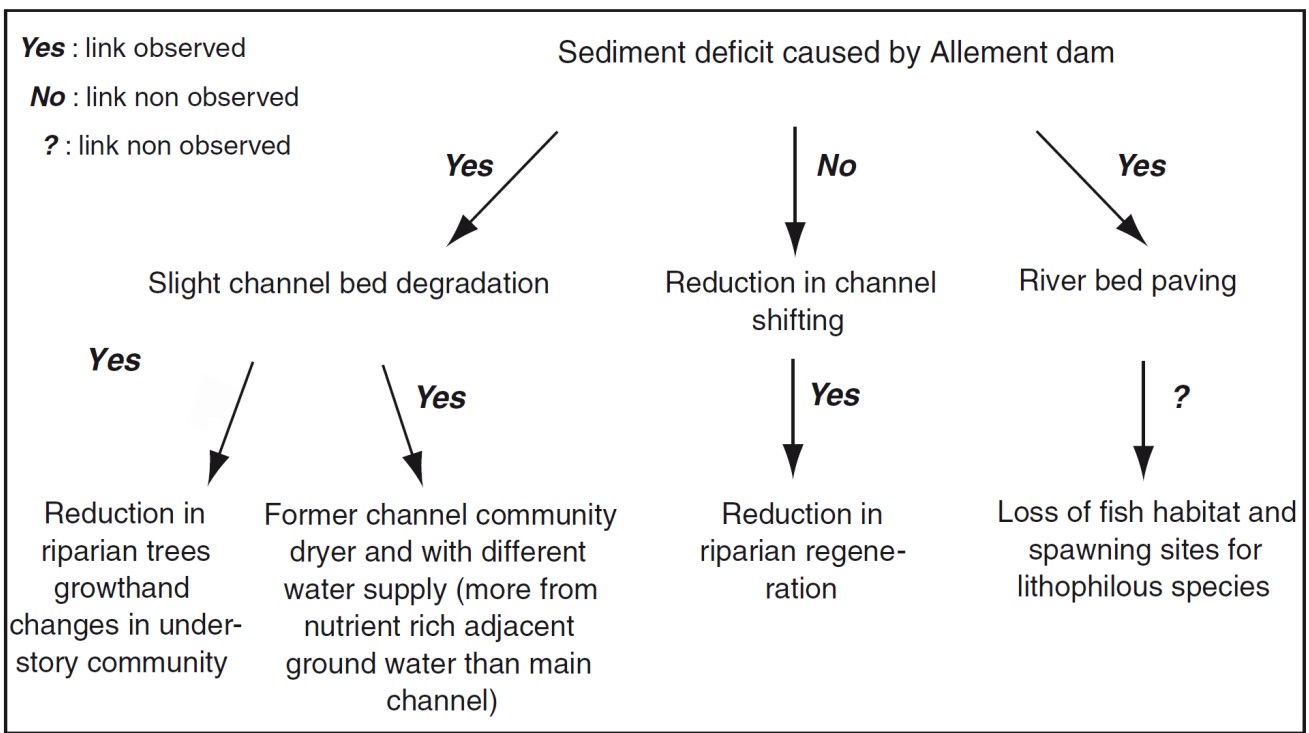

**Figure 2.** Example of causal diagnosis on the Ain River: summarising of the observed and non-observed links between sediment starvation, geomorphic adjustments, and ecological consequences (from Rollet et al. 2013 [36]; reproduced with permission of J. Wiley, 2023).

Sediment budgeting is a key aspect of morphological diagnosis. To identify effective rehabilitation measures, we need to have a clear idea of the orders of magnitude of sediment transport, the distance and timing of change propagation, and the contribution of the different basin compartments (e.g., tributaries versus floodplain). The sediment budgeting approach aims at estimating sediment erosion, transfer, and deposition through a given river reach by considering three-dimensional morphological changes [38,39]. It can be based on different techniques that may be more or less advanced, such as use of a bedload transport formula at a station [40], morphodynamic modelling [41], field validations based on sediment tracking [42], and sediment flux estimates based on geophones or traps [43]. This sediment budgeting approach is usually combined with historical information (topographical and planimetric) and field campaigns (bank height estimates, overbank fine sediment volumes), and sometimes with LiDAR-derived digital elevation models (DEM) to detect geomorphic changes following gravel mining or other human pressures on sediment storage and floodplain–channel interactions. For example, the work performed by Boutault [44] on the Dordogne River showed that since 1948 the river channel has undergone an overall deficit of 5.6 million m$^3$ of sediment; the 8.8 million m$^3$ of sediment extracted was partly compensated by bank erosion (3.7 million m$^3$ of sediment was

introduced), but also exacerbated by no upstream entrance due to damming and a downstream output of 0.5 million m$^3$ of sediment. It should also be noted that 20.5 million m$^3$ of sediment was also stored within the floodplain through an active process of afforestation and channel metamorphosis (from a wandering to sinuous single-bed channel) due to very significant peak flow lowering. This sediment budget provided information that was critical for estimating the potential effects of rehabilitation measures such as bank erosion promotion and the raising of peak flows, as well as for quantifying the volume needed to reach a certain improvement level.

## 4. Pre-Project Assessment: Evaluate Risks, Gains, and Potential Channel Responsiveness before Acting

Although rehabilitation projects are now becoming more frequent, a systematic framework for designing and monitoring rehabilitation actions is still lacking [17,45,46]. Rehabilitation actions remain experimental manipulations of local features of a river, sometime whole river reach, and are associated with uncertainties and risks that must be addressed in the early stages of projects [47,48]. In the case of sand deposition in the by-passed section of a gravel-bed river (Selves River, France; [49]), experimental (eco)morphogenic flows (e.g., the geomorphic domain of environmental flows; [3]) were designed to identify the most appropriate discharge to flush sand downstream without remobilizing coarser particles and impacting existing habitats. During three tests at maximum discharge (10, 15, and 20 m$^3$/s), each carried out for four hours, sand movement was measured using coloured tracers, sediment flux was surveyed using a Helley–Smith sampler, and repeated field measurements of bedforms were made. These experiments provided information on the best flow level and duration, facilitating the design of repeated measures to maintain the improved habitats and promote an adaptive approach to mitigate impacts caused by the dam.

In the case of artificial gravel augmentation oriented towards mitigating the adverse effects of sediment starvation downstream from dams, the risks include flood-stage rising in the area of the stockpile deposit, uncontrolled bed scour with the reactivation of bed-load transport, and threats to downstream infrastructures that depend on the sediment transfer pace. The challenge is thus to develop a knowledge feedback system through consolidation of the different stages of the rehabilitation project, thereby reducing the uncertainties relating to the expected benefits and risks before acting [42]. In situ and ex situ experiments can be performed to improve the rehabilitation design. The volumes of introduced gravels are generally selected on the basis of the mean annual transport capacity of the reach. Location, geometrical configuration, and grain size of the introduced gravel are also critical issues. In addition to in situ monitoring, feedback, often coming from pebble tracking [50–52], flume experiments [53–55], and numerical modelling [56,57], is developed to evaluate these issues, assess channel sensitivity, anticipate potential failures, and thus improve the rehabilitation strategy. For example, using a flume, Koll and Koll (2012) [58] tested the effects of different amounts of gravel augmentation on the stability of a static armour layer and associated sediment transport velocities prior to in situ gravel augmentation in the Rhine River (Figure 3). Among the uncertainties associated with gravel augmentation is the question of the availability of sufficient sediment volumes (as stocks or active production) to ensure the sustainability of the actions; the volume of coarse sediments stored in the floodplain is determined by the morphological sediment budget, and therefore the sustainability of gravel augmentation measures can be deduced from it. In a study on the Ain River, Talaska et al. (2014) [59] estimated from the sediment budget that the floodplain gravel supply should be available for 40 to 90 years to counteract the 15,000 m$^3 \cdot$year$^{-1}$ sediment starvation affecting the channel.

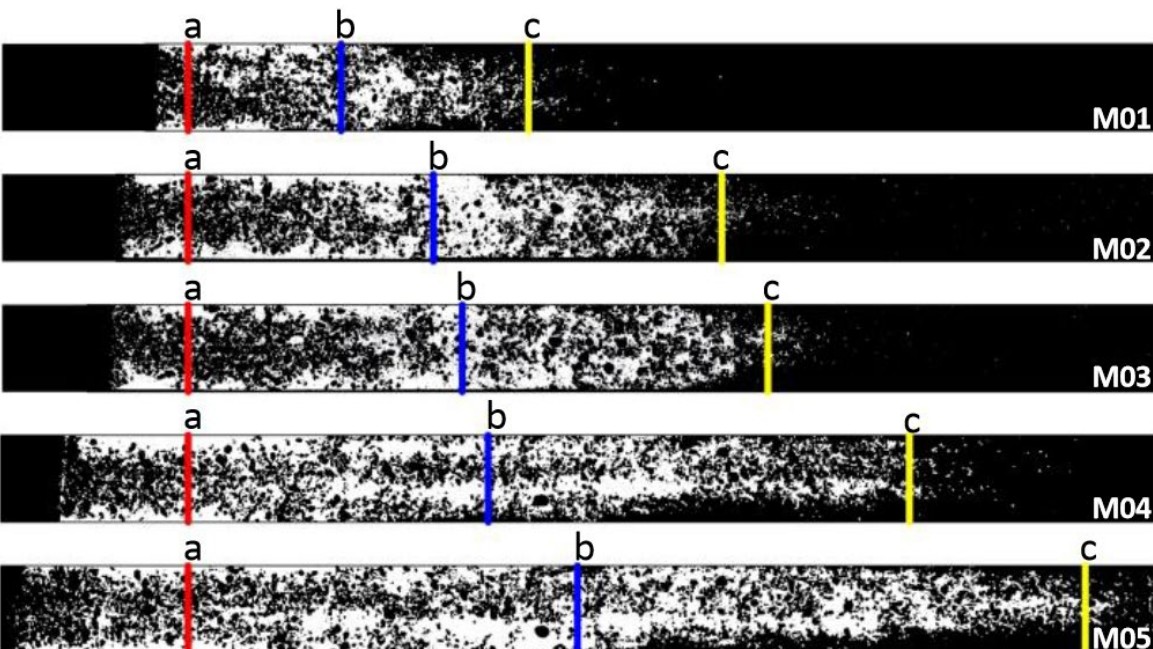

**Figure 3.** Flume experiment prior to the in situ gravel augmentation on the Rhine River: distribution of tracers after a run time of T = 30 min according to the tracer amount (M01 to M05: 1.1 to 4.4 kg), with a = front of the initial tracer bar, b = position of the balance point, c = position of the tracer front (6 m long, 0.3 m wide, and 0.4 m high flume) (from [58] Koll, K. and Koll, K. *Influence of Depot Size on Bed Load Transport Velocity over Static Armour Layers. In River Flow*; R.E. Murillo Munoz: London, 2012; Vol. 1, pp. 451–456; reproduced by permission of Taylor and Francis Group).

The geomorphologist's perspective contributes to providing an inference on how physical compartments can react in the future, and the implications for habitat improvement and sustainability. Geomorphological analysis is a way to evaluate not only potential risks, but also potential gains. At this stage, collaboration with biologists is needed to assess potential ecological responses to manipulations of forms and processes. The main ecological objectives of the gravel augmentation on the Rhine River were a gain in substrate diversity to improve spawning habitat for lithophilic fish species and enhance the recruitment of pioneer plants, and a gain in surficial and sub-surficial water exchanges for the associated benthic and hyporheic diversity [42]. During the initial phase of pre-project assessment, ecological monitoring was proposed to evaluate the potential gain in such specific measures because of the potential cumulative effects of several rehabilitation measures (gravel augmentation, instream flow increase) and environmental drivers (water quality, colonization by invasive species) on the biological communities. However, the combination of before–after and control–impact monitoring approaches allowed differentiation of the effects of the rehabilitation measures and environmental drivers on the richness of the macrophyte species, recruitment of pioneer terrestrial species, and taxonomic richness of macroinvertebrates [60]. On the basis of these results, a conceptual model of key cause–effect relationships and interactions was developed to describe positive, negative, and mixed effects of rehabilitation, with this model providing a design framework for maximizing ecological gains in the more extended rehabilitation project (Figure 4).

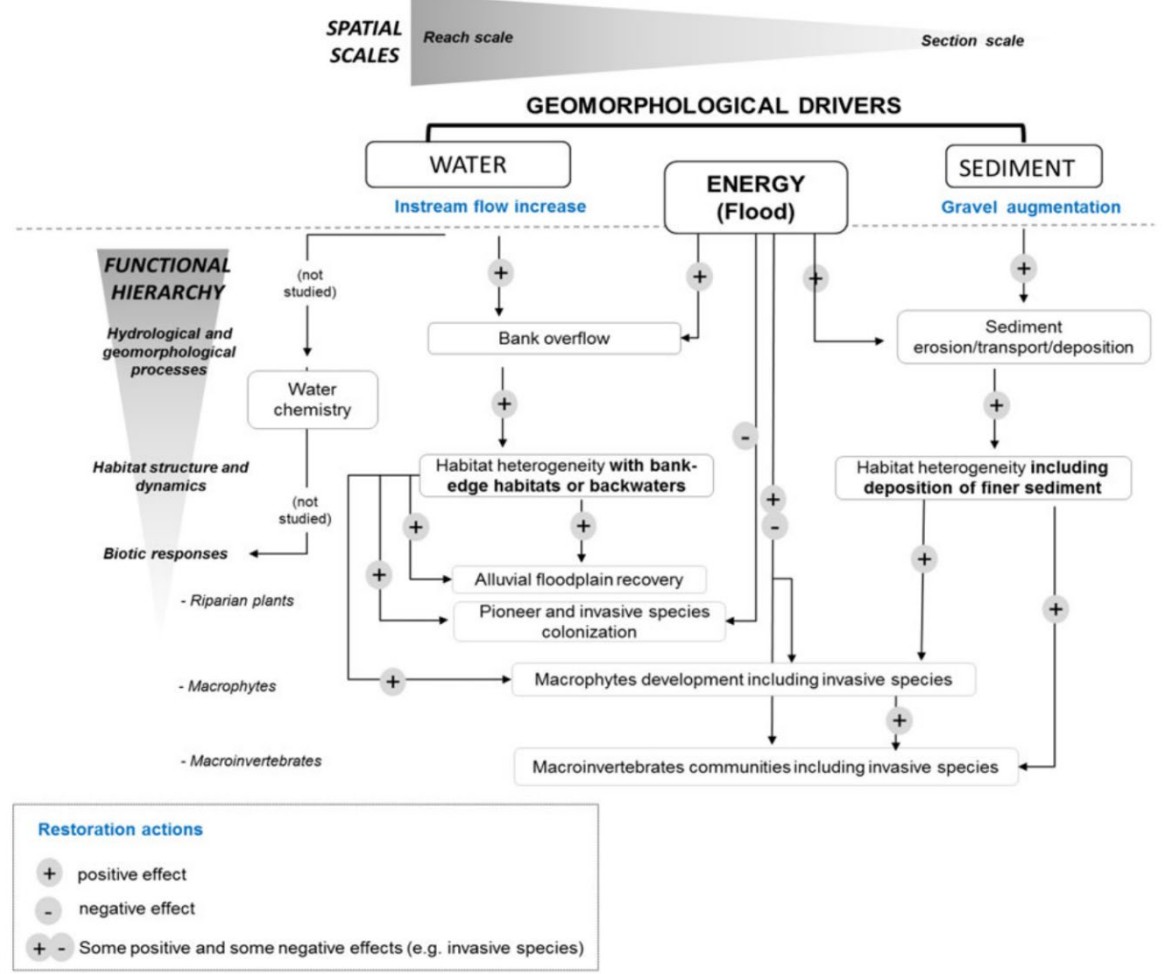

**Figure 4.** Conceptual model based on the ecological monitoring of the Rhine River, illustrating the relationships/interactions between compartments of the river system and the two types of rehabilitation action (Staentzel et al., 2018 [60]; reproduced by permission of J. Wiley, 2023).

## 5. Post-Rehabilitation Project Assessment: Evaluate Success and Provide Predictive Models for Rehabilitation Design

Because rehabilitation can fail, monitoring is often recommended to assess its success or failure, to allow reporting of it and adaptation of existing measures with additional ones. Monitoring is even more important when a pre-project assessment step to evaluate potential risks and gains is not implemented. Although ecological indicators are most often used, morphological ones can also be considered for assessing the sustainability of measures, being taken as indicators of rehabilitation success. Morphological indicators are also monitored in combination with ecological ones to help interpret ecological responses and better understand any additional measures required, as well as to provide predictive models that can serve as pre-rehabilitation tools useful for improving design efforts to achieve optimal response scenarios. In some cases, morphological assessments can also be cheaper than ecological monitoring, and the diversity of forms can be viewed as a surrogate for the diversity of communities or expected functional responses.

### 5.1. Is Rehabilitation a Success? What Can a Monitoring Framework Tell Us?

We here provide three examples to illustrate how to evaluate the success of rehabilitation using morphological approaches. The first two are based on a before/after survey design, whereas the third illustrates a control/impact strategy.

The first example is based on a before/after survey design that considered both the sustainability of measures and habitat improvement. One of the key measures to improving the ecological conditions of the Rhône River was to rehabilitate the side channels in by-passed reaches. To maintain these critical habitats that the river can no longer sustain by itself, a strategy based on mechanical rejuvenation by dredging and/or partial-to-full reconnection of the extremities of former channels with the main river has been implemented since 1999. The guiding principles of the rehabilitation were to improve ecological functioning and maximize the diversity of habitat conditions in side-channels at the reach scale in order to maintain a wide range of successional stages [61]. The main objective of the monitoring was to assess the relevance of the rehabilitation strategy in terms of efficiency and sustainability on the basis of a before/after survey design [62]. Indeed, morphological adjustments to former channels are highly site-dependant, and can be strongly influenced by changes in geometry caused by rehabilitation works, and the use of control sites in such environments is therefore complicated. A quick, simple, and easily-reproducible protocol to survey a large number of side channels was developed. For each channel, this consisted of probing water depths and fine sediment thickness along the centreline of the waterbody and sampling surficial fine sediment for grain size analysis. These measurements were carried out at least once before rehabilitation and every two years on average after rehabilitation. They are complemented by a continuous monitoring of the flood regime as a first-order control parameter of the hydromorphological evolutional trajectory (duration, frequency, and intensity of flood pulses). After more than a decade of monitoring, the results showed that the hydromorphological conditions observed after rehabilitation were often very similar to those observed prior to the work, with the exception of active secondary channels that were reconnected at both ends [63,64]. In most cases, the geometry of side channels was modified by rehabilitation without substantially modifying the main processes controlling the conditions (i.e., connectivity between side channels and the main river channel). Indeed, the geometries of upstream alluvial plugs of the backwater channels (permanently connected with the river only at their downstream end) were not modified. The excavation of backwater channels led to an increase in their ability to trap diffused sediments during backflow events. This trend is all the more obvious because most channels were perched above the main river channel before rehabilitation. In terms of potential life span of the rehabilitated channels, the increase in water depths resulting from rehabilitation largely compensated for the higher fine sediment accumulation rates observed after rehabilitation.

Our second illustration of the before/after monitoring framework focuses on gravel augmentation contexts, for which the technologies of radio frequency identification (RFID) have become an almost mandatory technique for validating the effectiveness of measures. Along with topographic surveys, these technologies are frequently applied to survey the mobility of the coarse fraction (b-axis > 20 mm) of bed sediment in rehabilitated sections [42,51,52,65–67]. This approach is also used for monitoring the morphologic effect of river rehabilitations and assessing their performance. Classically, passive low frequency transponders (so called PIT tags) were used because their small size, low cost, and long operating life made them ideal for long-term monitoring of large sets of tracers [42,66–68]. These PIT tags have recently been complemented with the addition of active ultra-high frequency transponders (so called a-UHF tags [69]). Although more expensive and with a shorter operating life, the latter do not suffer from signal collision and have a wider sensing range (up to 40 meters) allowing faster surveys with a variety of protocols, depending on study purposes and available resources [70]. They have even been used to monitor streams exhibiting intense vertical bedload mixing and large transport distances [52]. Moreover, they offer the possibility to determine active layer thickness and the displacement of constrained particles during the first survey after injection when deployed in columns into the active layer of wadable streams [44,71]. The combination of these two technologies offers new perspectives for long-term monitoring and sediment transport budgeting. When monitoring rehabilitation projects, RFID technologies are helpful to:

1. Validate the grain size of the replenished sediment mass that can be effectively transported by the stream once rehabilitated, and which is estimated from hydraulics models and flow records;
2. Estimate the effects of the rehabilitation on aquatic habitats in terms of improved bed surface area and sustainability;
3. Estimate, improve, and validate the volumes needed and the necessary frequency for reinjection operations.

The third example is illustrated by a control/impact design implemented on the Drac River following a large-scale rehabilitation scheme. The Drac River is a braided Alpine river with active bedload transport and a mean annual discharge of 9 m$^3$/s. In its section upstream from *Saint-Bonnet-en-Champsaur*, the river suffered from significant incision that reached the underlying clay layers and from a clear change in planform style from braided to single-threaded river due to intense mining in several places up until 2012 [72]. Following this degradation, 3.9 km of river and an area of 27 ha were rehabilitated, with channel widening and clearcutting of riparian vegetation to re-open a stabilised landscape, rehabilitation of local wetlands within the floodplain, and an artificial gravel augmentation (+450,000 m$^3$). This induced an average mean bed elevation rise of +3 m, a channel width increase from 30–40 to 80–120 m, the recovery of braided patterns, and partial restoration of bedload transport dynamics [72,73]. However, one of the expected outcomes from these restored braided patterns was the provision of important functional processes such as groundwater exchanges and cold spots, which are critical components of such systems, allowing it to be considered as an Alpine reference [74]. Thermal behaviour related to channel geomorphology was then assessed using airborne thermal infrared (TIR) imagery within a control–impact strategy, comparing the rehabilitated reach with an upstream natural braided reach (Table 1). Airborne TIR now has long history of use to study rivers [75] as it allows assessing of surface temperature with high precision and high resolution over large spatial scales [76]. It has been successfully used to characterise the thermal diversity of a range of gravel-bed rivers [77] but is rarely used within a post-rehabilitation monitoring strategy.

**Table 1.** General characteristics of the control and rehabilitated sections of the Drac River (* braiding index = Pttw (see Figure 2; [73,78]); normalised bed relief index = BRI * [73]).

|  | **Control Section** | **Rehabilitated Section** |
|---|---|---|
| Thickness of alluvial material | 10–15 m | ~3 m on average |
| Length of section | 3.9 km | 4.0 km |
| Average width (±1 S.D.) | 78 ± 27 m | 80 ± 30 m |
| Average slope | 0.0092% | 0.0107% |
| Braiding index * | 4.14 | 4.35 |
| Wetted width vs. active width ratio (R) | 0.24 | 0.49 |
| Normalised bed relief index * | 0.003 | 0.003 |

From a morphological perspective, the rehabilitation has proven to be effective; active width has increased, the braiding index has recovered, and some bedload transport has been reactivated (Table 1). However, from a temperature perspective, the behaviour of the reach remains different from that of the control reach: the thermal gradient is positive (+0.52 °C/km), whereas it is negative in the control reach, and the density and diversity of thermal habitats are low (Figure 5a,b). Several lateral seeps were identified along the floodplain margins as well as poorly-connected cold side-channels, showing that cold-water inputs exist but connectivity flow paths are yet to be restored. Similarly, a limited number of hyporheic upwellings were found in the rehabilitated reach, showing that vertical connectivity is also low. The limited signs of restored thermal functions can be explained by several factors [79]. First, the layer of available alluvium is relatively thin

(Table 1) and too 'young' to actively interact with the groundwater table, as even internal recycling of surface water along gravel features is limited. Despite the recovery of braided patterns, no morphogenic discharge occurred following the rehabilitation to re-work the sediment. Geomorphic features have remained rather artificial (being the ones designed by the operator), with, for example, a wide wetted channel that is much higher than the one of the reference reach (Table 1), and which causes the water to be spread thinly over a large area, and thus to be more exposed and vulnerable to solar radiation.

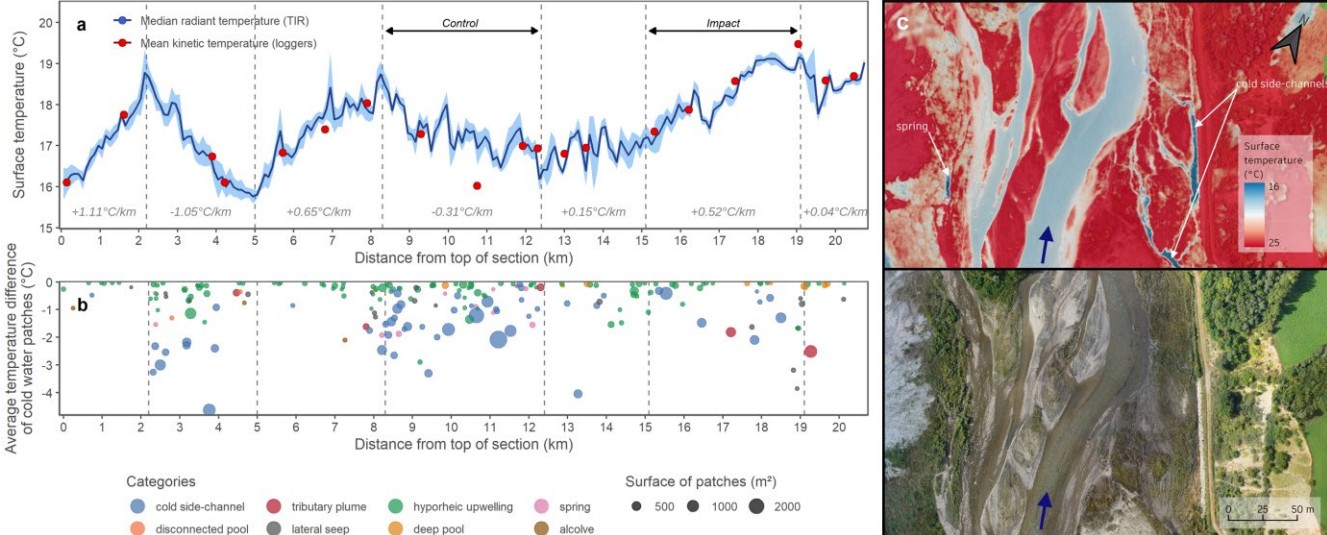

**Figure 5.** Thermal responses of a rehabilitated reach in comparison with a control reach following rehabilitation of the upper Drac River, France. (**a**) Longitudinal temperature profile with local thermal gradients, (**b**) distribution, types, and characteristics of cold-water patches, (**c**) TIR image and orthophoto of a section of the rehabilitated reach illustrating the disconnection between cold-water features and the main overly-wide restored braids (modified from Marteau et al., 2022 [79], reproduced by permission of J. Wiley and Sons, 2023).

This case study (among others, see [79]) showed that a successful morphological rehabilitation does not necessarily mean a successful recovery of functions. Some functions (i.e., those related to temperature in this example) are more complex and/or slower in their responses to changes (here assumed to be positive changes) than are morphological responses, and therefore so are the timeframes of most monitoring programmes implemented as part of the rehabilitation design. This example also suggests that alternative tools (e.g., airborne TIR) providing alternative indicators (e.g., thermal gradients, cold-water habitats) can be used to help refine the assessment of the success or failure of rehabilitation schemes, and also shed new light on some of the reasons for observed failures.

The successfulness of rehabilitation is commonly assessed using biological indicators based on similar control/target monitoring frameworks. However, as shown before, success is sometimes only partial, or even completely absent, for various reasons (see Section 2). In such contexts, a better geomorphic understanding can help promote the additional or correcting measures needed to improve the initial rehabilitation scenario. The effects of gravel bar redesign on plant community functional composition were assessed along the Rhône River by [80]. They compared bar elevation and soil texture gradients on bars being newly reprofiled in sediment starved reaches (target) with bars located in a reach where they can be naturally rejuvenated by sediment transport (control). The co-occurrence of species with contrasting traits was higher in highly disturbed environments, demonstrating the importance of rejuvenation processes in inducing ecological improvement related to sediment transport. The monitoring showed that the measures implemented were not sustainable because the initial design (bar reprofiling) was not process-based. To promote an effective ecological rehabilitation of such riparian zones, the rehabilitation of bedload

transport was recommended instead. This was presented as a more effective measure because it provides more natural disturbances that allow diverse and repeatedly renewed vegetation assemblages to develop, with the aim of preventing biodiversity decline through time.

### 5.2. The Feedback Loop of Post-Rehabilitation Monitoring Data: Feeding Predictive Response Models

Post-rehabilitation monitoring data are useful for assessing success, but can also be used to feed models that can then provide useful tools for improving practices and guiding future rehabilitation designs. Monitoring is essential to increase operational feedback and provide models for a priori use in the targeting of rehabilitation actions [81].

Models based on a 15-year monitoring of rehabilitated side channels along the Rhône River provide a good example of such findings. It was demonstrated that grain size patterns and associated habitat types [63], fine sediment accumulation rates, and the life spans of rehabilitated channels as aquatic habitats [64], can be successfully predicted using simple metrics describing the hydrodynamic functioning of these channels: the frequency and magnitude (i.e., maximum total boundary shear stress) of upstream overflow events and the maximum intensity of backflow events (i.e., maximum backflow capacity) (Figure 6).

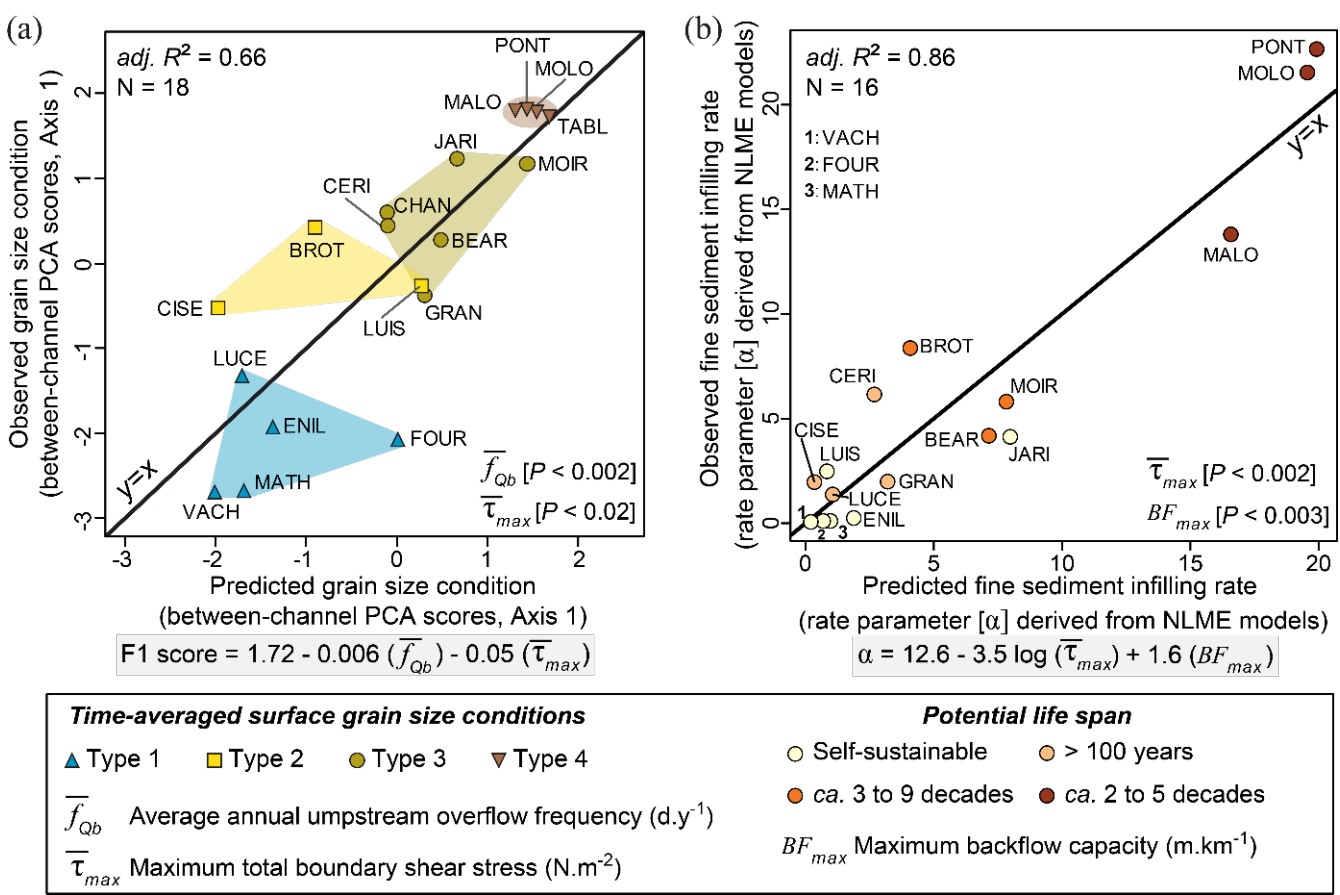

**Figure 6.** Example of empirical statistical relationships allowing the prediction of (**a**) time-averaged surface grain size conditions and associated types, and (**b**) fine sediment infilling rates in rehabilitated side channels of the Rhône River, from the frequency and magnitude (i.e., maximum shear stress) of upstream overflow events and the maximum intensity of backflow events (i.e., maximum backflow capacity) (modified and adapted from [63,64], respectively). PCA, principal components analysis; NLME, nonlinear mixed-effects (NLME) models. Reproduced by permission of Elsevier (**a**) and J. Wiley and Sons (**b**).

The main interest in these predictors lies in the fact that they reflect the control exerted by the geometry of side channels (e.g., the morphology of the upstream alluvial plug, slope conditions), which managers can target or modify. Such statistical empirical relationships make it possible to quantify how engineering decisions relating to the design of channels can influence their hydromorphological adjustment after rehabilitation. On this basis, Marle et al. (2021) [82] combined calculation of the changes in the side channel morphology driving changes in local hydrology (upstream overflow and backflow processes) with statistical modelling of aquatic macroinvertebrate occurrences in response to these hydrological metrics. This eco-morphological modelling framework allowed prediction of changes in macroinvertebrate diversity along successional sequences in side channels according to different rehabilitation scenarios. All these models represent relevant operational tools to guide future project design, and can be used, for example, to predict the likely effects, ecological efficiency, and sustainability of side channel rehabilitations, or to target physical habitats that are infrequent or missing at the reach scale. Such approaches provide a good illustration of the gradual transition from trial-and-error attempts to rehabilitation works based on the science of design (sensu [83]).

Another illustrative example deals with particle mobility within a gravel augmentation scenario using the RFID technologies and monitoring data introduced in Section 5.1. Two main processes can be tracked using RFID-tagged stones: (1) the downstream migration (advection) of particles [84], which is the distance of transport of the tracers' cloud centroid; and (2) the particle dispersion (diffusion), which evaluates the variance of the distance of transport of the tracer population [85–87]. The information derived from particle tracking experiments can thus be used to feed models linking gravel transport to different hydraulic parameters [88], e.g., cumulative excess stream power and flow duration. These models are usually based on surveys accomplished over a long time-span in order to record a large enough range of flow and transport conditions for a single site, and/or use data collected at several sites that are adequately normalized and are compared together to provide more robust generic models. In this regard, by combining a probabilistic function for downstream tracer dispersion with a one-dimensional sediment transport model based on field observations using RFID-tagged stones, Vazquez-Tarrio et al. (2023) [89] were able to estimate the time needed to export all the augmented sediment from a by-passed reach of the Rhône (Figure 7) and to simulate different scenarios of gravel augmentation. The model predicted that even 50 years after the injection of gravels, the sediment will remain stocked in the by-passed channel, and almost no sediment will arrive at the downstream non diverted-channel (Figure 7b,c). This approach is very useful for evaluating the amount of gravel required for a potential geomorphic diversification of the channel features without causing disruption to human uses such as downstream navigation. It can help to identify, before any rehabilitation actions, the best locations, frequencies, grain size, and volumes of injection, and the potential areas where we should expect significant habitat improvements and major discontinuities in sediment transport that must be considered.

One of the main questions commonly addressed in river rehabilitation is the required duration of post-restoration monitoring to properly evaluate operation success. The answer to this question differs according to the biological or physical compartments targeted by restoration, as well as the restoration measure itself. For example, in the case of dam removal, the recovery of longitudinal fluxes (i.e., flow, sediment transport, biological organisms) is almost immediate [90,91]. However, the recovery of biophysical processes derived from the restoration may require years or even decades and may be more difficult to detect (e.g., adjustment of physical habitats, stabilization of the new fluvial landscape, changes in the composition of biological communities) [13]). In particular, the responsiveness of geomorphic features to river restoration is closely linked to flood frequency and magnitude, upstream sediment supply, and vegetation recruitment and growth processes that influence riverbed roughness. This means that post-restoration monitoring must be fairly long, at least a decade (or including the occurrence of a $Q_{10}$ flood) to be able to evaluate channel adjustments, and potentially two decades to accurately assess the sustainability

of the measures and evaluate the validity of predictive models. In the case of the Rhône side channels rehabilitation programme, a 10-year period was found to be a minimum to determine whether sedimentation in restored side channels was time-dependant, or whether these could be reset by scouring during large floods [64]. Similarly, the recent gravel augmentation operations conducted in by-passed channels of the Rhône and the modelling strategy developed to predict the kinematics of gravel propagation [89], also show that 10 years is the minimum temporal extent to accurately validate the effectiveness of the restoration measure (Figure 7). In the case of the Drac River, it seems that restoration is not sustained after 10 years and an additional monitoring period that includes a large (e.g., $Q_{10}$) flood would be required to validate the initial diagnosis (Figure 5). Finally, beyond the duration of the monitoring, defining the best temporal frequency of observations is also critical. According to targeted biophysical indicators and expected responses related to river characteristics (e.g., hydrological regime, sediment supply, basin topography), surveys can be performed once a year or every two to three years. Planform-based indicators from satellite imagery, in the case of river features greater than about 100 m$^2$, enable almost real-time monitoring with Sentinel-2 images (10 m in resolution) or nanosatellite images (about 4 m in resolution for planet.com) (see next section and Figure 8).

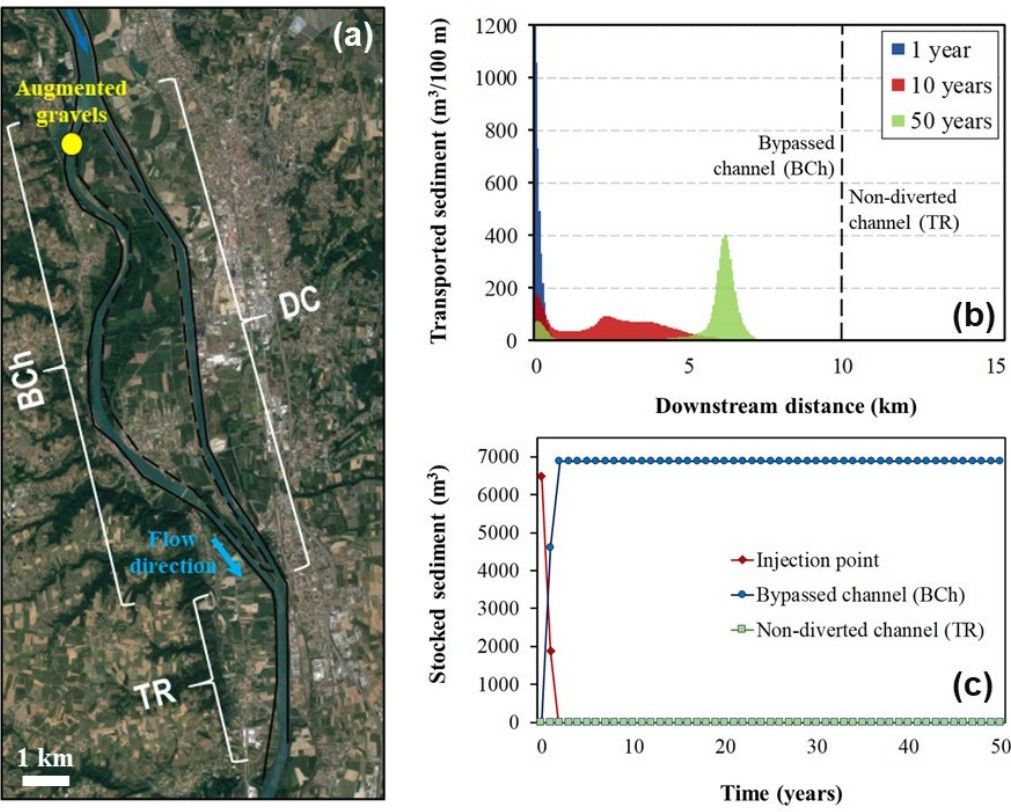

**Figure 7.** Temporal evolution of the stock of sediment (**b**,**c**) introduced in the by-passed section of Péage-de-Roussillon on the Rhône downstream of Lyon (**a**) based on a modelling framework using dispersion probability and 1D transport capacity. BCh: bypassed Rhône channel. DC: diversion canal. TR: total (non-diverted) Rhône channel.

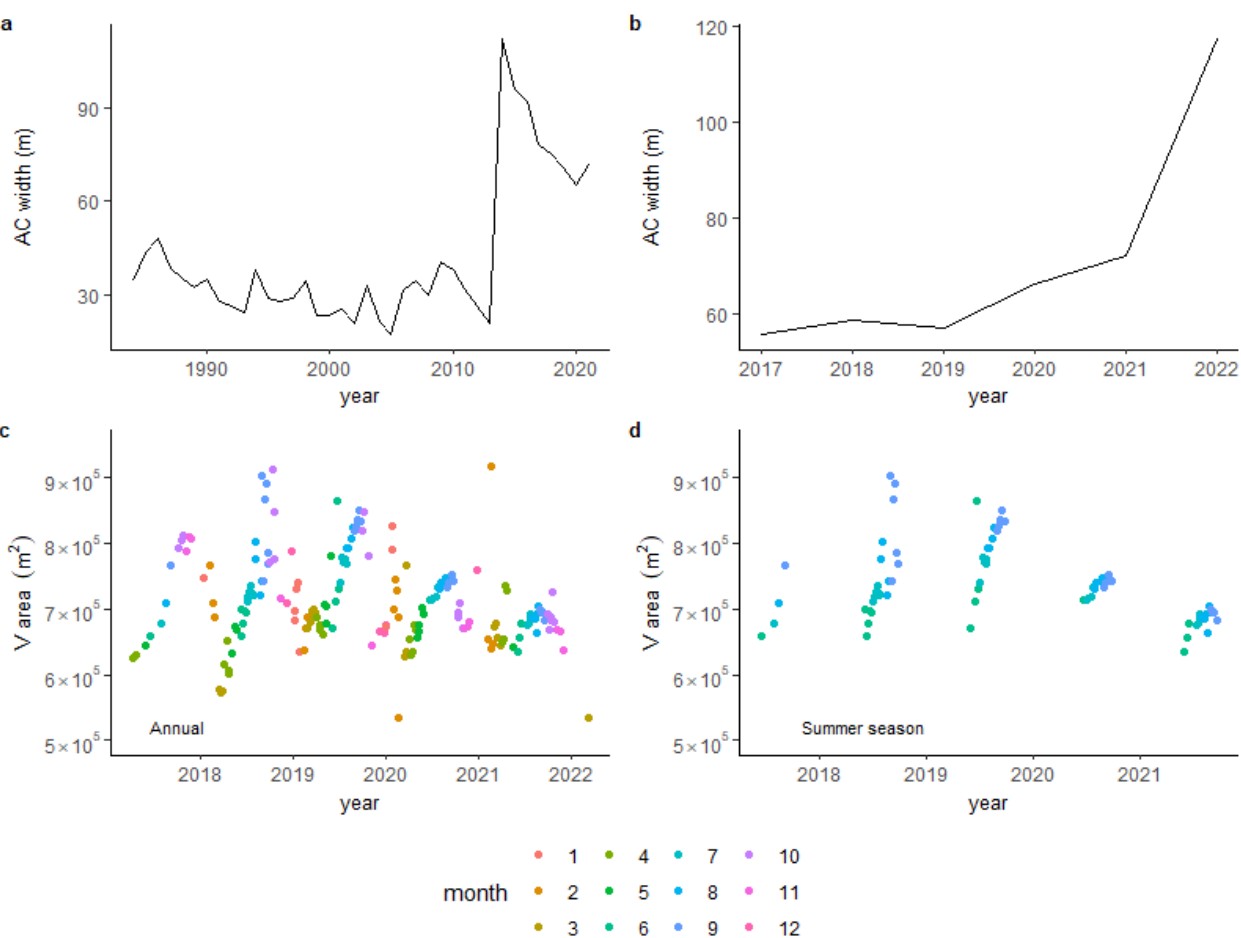

**Figure 8.** Temporal evolution of active channel width and riparian vegetation encroachment of the Drac restored reach, based on spectral indices detected from satellite remote sensing: (**a**) from Landsat archives (before and after restoration works of 2013–2014; 30 m resolution), (**b–d**) from Sentinel-2 imagery (3 to 8 years after restoration; 10 m resolution). AC: active channel; V: vegetation.

### 5.3. A New Monitoring Era Is Emerging with Remote Sensing Information

Recently, Fryirs et al. (2019) [92] wrote about the ongoing revolution in monitoring and measurement techniques that is transforming data availability and approaches within earth and environmental sciences. The recent increase in data availability (particularly remotely-sensed data) offers perspectives on the generalisation of large-scale analyses of rivers and their catchments [93,94]. Advances in acquisition techniques allow higher resolution and higher frequency data to be quickly gained with archives allowing us to explore changes occurring over the last four decades and, more recently, almost on a weekly basis. These new data enable geomorphologists to obtain information that is relevant at each of the scales of interest and to track indicators over time, including during the diagnosis phase and the pre- and post-action assessments.

For example, Figure 8a,b shows the evolution of the annual active channel width (i.e., including sediment and wetted areas) along the Drac River obtained from spectral indices (MNDWI, NDWI, and NDVI) [95] through the retrieval of 309 Landsat (30 m resolution) and 72 Sentinel-2 (10 m resolution) images acquired during the summer season (i.e., to include the vegetation extent). This clearly displays that the active channel (AC) width progressively decreased since the 1980s to adjust to impacted conditions until the restoration operations in 2013–2014, when a significant increase in AC width was observed, followed by a further adjustment during the following years. From 2017 onwards, Sentinel-2 images allowed assessing of the detailed sequence of adjustments until the recent increase in 2020–2021, as a consequence of additional gravel augmentation operations combined with

the occurrence of large floods. Additionally, Figure 8c,d illustrates the annual vegetation encroachment (in terms of vegetated area increase) within the total active channel from the analysis of all 184 Sentinel-2 archived images in the area (c) and the 72 Sentinel-2 images over the summer season. These examples demonstrate how our capacity to directly measure and monitor geomorphic characteristics is significantly enhanced. This allows for better orientation of field surveys as well as calibration and validation of models, which in turn leads to positive feedback for developing models at larger scales [96].

Today, a strong challenge lies in harmonizing and sharing more widely and effectively datasets and methods [94,96]. This is particularly important to improve communication with managers and decision makers [93]. There is also a growing number of regional 'observatories' that monitor river conditions and biodiversity, filling the gap between river scientists and river managers [94,97].

## 6. Targeted Rehabilitation Using Upscaled Geomorphology

Although river rehabilitation has been developing for over 20 years, practical difficulties in planning projects remain, despite the current incentive and institutional frameworks. Many rehabilitation projects were decided upon and implemented on an opportunistic basis; they started as experiments from which much has been learnt, both in terms of success and failure [98]. Following this initial period within which river rehabilitation emerged, there is a current call for river managers to move from opportunistic to strategic projects. Upscaled geomorphology supports rehabilitation planning and prioritization because it provides information for evaluating river conditions (e.g., levels of pressures and alterations, available room for a river within the valleys) and potential river responses to passive or active rehabilitation measures. This knowledge can help planners prioritize operations and identify potential benefits where a given rehabilitation measure should prove efficient or valuable.

Geomorphological understanding of river processes provides an organizing principle for long-term river management and improvement. As an example, Kline and Cahoon (2010) [99] reported how geomorphology has been incorporated within the Vermont State (USA) conservation strategy designed to create an effective long-term river management framework. In Europe, the Water Framework Directive adopted in 2000 set challenging river rehabilitation goals that have required a systematic mapping of hydromorphological conditions within European member states. Gaining a broader perspective allows for a better understanding of the context of (and more accurate) geomorphic interpretations of the landscape [93], because the characteristics of the watershed govern river processes. Jain et al. (2020) [100] demonstrated that lithological and topographical variability, which strongly influence the erosion rate, sediment supply, and channel shape at the catchment scale, play a major role in defining stream hydrogeomorphic characteristics. A large-scale analysis of these controls is therefore a means of predicting a river's characteristics. Knehtl et al. (2018) [101] found similar results in the prediction of habitat conditions between methods using remote sensing data and field survey data. The use of a nested hierarchical framework to improve geomorphological assessment is recommended by several authors (e.g., [102]), particularly concerning the impact of anthropogenic pressures. Multi-scale approaches can lead to the integrated understanding of fluvial processes that are necessary to assess the sensitivity of rivers to anthropogenic pressures and in turn enhance the effectiveness of rehabilitation projects. The integration of the concept of connectivity, particularly between sedimentary sources and rivers [103], is an important way to improve the effectiveness of these approaches. Schmitt et al. (2019) [104] used a combination of connectivity modelling and geomorphic regionalization to determine mean sediment yield and estimate the impact of future dam construction on sediment fluxes. This kind of contribution provided stakeholders with a strategic vision at the basin scale and opened the way to a more sustainable development of hydro-power in the Mekong basin. Diagnostic tools, such as the River Style Framework [102,105], are now widely used to forecast a part of the geomorphic response in multi-scale analyses. Using this framework, Marçal

et al. (2017) [106] were able to produce a geomorphic understanding of several Brazilian rivers at the catchment scale, which served as a basis for forecasting future adjustments for different management scenarios. Such diagnostic tools can also be used as management tools to gain an understanding of the geomorphic trajectories of streams and to forecast the effectiveness of rehabilitation projects [94]. Bizzi et al. (2019) [107] provided an example of an assessment of potential human-induced alterations at the network scale based on low-resolution regional LiDAR data combined with aerial photographs and hydraulic scaling laws from channel geometry. They were able to determine a range of conditions of channel depth and width observed in the Po River network of the Piedmont region in Italy, and to compare them with references. Wheaton et al. (2018) [108] upscaled site-scale ecohydraulic models to provide information for population-level salmonid life cycle restoration within the Columbia basin (Figure 9). They characterized each site's geomorphic reach type, habitat condition, geomorphic unit assemblage, primary production potential, and thermal regime, and produced drainage network-scale models to estimate these same parameters from coarser remotely-sensed data available across entire populations within the Columbia River Basin. In a further example, Alber and Piégay (2017) [109] assessed potential lateral erosion using stream power, land-use maps, and a potential sediment supply map estimated from the local maximum active channel width. From such analysis, it is possible to identify reaches that are still potentially responsive within alluvial valleys/contexts where lateral shifting is not hindered or may be restored. Such combinations of data and models provide practical information for targeting actions and selecting sites that are good candidates for process-based conservation or other forms of intervention such as rehabilitation.

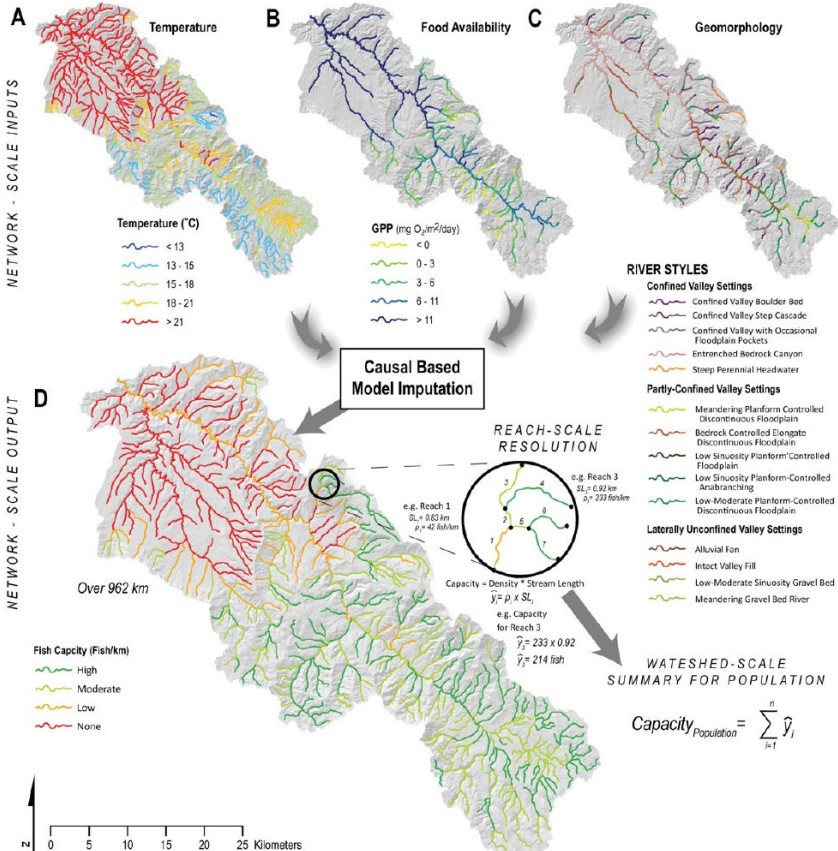

**Figure 9.** Illustrative example of upscale approaches showing network-scale inputs of temperature (**A**), gross primary production (**B**); i.e., proxy for food, and river styles (**C**) used to inform a model imputation to drive a network-scale fish response output (**D**) within a sub-basin of the Columbia River basin (from Wheaton et al. 2018 [108], reproduced by permission of J. Wiley and Sons, 2023).

In conclusion, the use of a multi-scale framework improves the understanding and interpretation of geomorphological structures and processes, notably by integrating sediment transfer processes, network connectivity, and existing (or future) pressures. The availability of remote sensing data allows these analyses to be performed in a more automated manner, at the most relevant scales, and from anywhere. The upscaling of geomorphology therefore offers managers the possibility to better target where to act, and to prioritize actions to be carried out according to their potential success by considering different evolution scenarios. By using these types of tools, real integrated management/rehabilitation strategies at the watershed scale are now at hand.

## 7. Conclusions

This contribution provides a series of examples illustrating how geomorphology can be used to maximize river rehabilitation success. Knowledge can be theoretical or practical, at both local and large scales. Geomorphology is part of the emerging environmental engineering that is applied to river conservation, mitigation, and rehabilitation. It is needed to promote sustainable river management and nature-based solutions founded on process-based understanding of systems. There is a need to clearly establish robust diagnoses based on river trajectory and river capacity to adjust to changing conditions. Such responsiveness is important for identifying the best rehabilitation measures over a range of solutions, from active to passive. If the potential contribution of geomorphology to rehabilitation is now demonstrated, the practical field is evolving very quickly thanks to technological developments (e.g., new sensors for sediment transport monitoring; drone imaging) and improved/generalized monitoring facilities that generate new continuous real-time data on river processes and conditions. These data offer new opportunities for the development and validation of physical response models, and even ecological response models. Within this favourable context, upscaled geomorphology emerges as a new critical research and application field: the integration of new data fluxes, such as regional LiDAR and images with increasing temporal and spatial resolution, requires new algorithms, computer interfaces, and data platforms that support semi-automated expert-driven river analyses and interpretations that include field insights. Tight integration between scientific and operational concerns on data usage should be regarded as a condition required to 'advance our understanding of river systems, translate information into knowledge, and raise the standards of river management practices' [92].

**Author Contributions:** Conceptualization, H.P.; Writing—original draft, H.P., F.A., M.C., B.M., J.R., C.R. and D.V.-T.; Writing—review & editing, H.P., B.B., F.A., M.C., B.M., J.R. and D.V.-T. All authors have read and agreed to the published version of the manuscript.

**Funding:** Labex DRIIHM French programme 'Investissements d'Avenir' (ANR-11-LABX-0010), which is managed by the ANR.

**Data Availability Statement:** Not applicable.

**Acknowledgments:** This study was conducted within the Rhône Sediment Observatory (OSR), a multi-partner research programme funded through the Plan Rhône by the European Regional Development Fund (ERDF), Agence de l'eau RMC, CNR, EDF, and three regional councils (Auvergne-Rhône-Alpes, PACA, and Occitanie). This work was cofounded by the Labex DRIIHM French programme 'Investissements d'Avenir' (ANR-11-LABX-0010), which is managed by the ANR. This work was performed within the framework of the EUR H2O'Lyon (ANR-17-EURE-0018) of Université de Lyon, within the programme 'Investissements d'Avenir' operated by the French National Research Agency (ANR). The authors kindly thank Karl Embleton for his proofreading work, the guest-editors, and three anonymous reviewers for their comments.

**Conflicts of Interest:** The authors declare no conflict of interest.

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
