# Peer review of "Why Consider Geomorphology in River Rehabilitation?"

_land, doi:10.3390/land12081491_

Round 1

Reviewer 1 Report

The manuscript describes and condenses the role and importance of geomorphology on river rehabilitation, considering that any action to recover the rivers can only achieve rehabilitation and can not be considered restoration due to the changes in the basin conditions and river functioning drivers.

It is a very good, interesting and useful manuscript, with examples of geomorphology role and application on rivers improvement.

Another question is a whole basin perspective of rehabilitation measures, which is mentioned on upscaled geomorphology section. Some of the diagnostic tools are very fieldwork and time consuming and alternative proposals based on remote sensing, LiDAR, aerial photographs tools are explained, however these, really interesting, are very difficult to be implemented in many small, narrow river networks

One of the aspects I think it should be addressed is the duration of the post intervention monitoring. This is a key issue that should be considered when doing the planification of the actions, and having and shared database of the results of rehabilitation actions.

It can be accepted in the present form after the correction of some formal and minor aspects:

- Figure 1. This figure extracted from Piégay et al 2016a, it is not well seen in the version I have seen, it is deformated, labels are not complete and the different levels of explanations are not in their correct site.

- Figure 3a and 3b, why are they in the same figure? In my opinion it is more clear if they are separated in two different figures, Figure 3 and Figure 4, respectively.

- Figure 6. Indicate flow direction and scale on (a). In (b) indicate the units in Downstream distance, also 10 and 5 years surface hide the previous ones, could be represent in a transparent way? In b and c, include the codes BCh, DC, TR when correspond. (c) the blue line means that the injected sediment on the injection point is completely wash out in just a year, from 6500 m3 to 0 m3? However, in the by-passed channel more than 5500 m3 are stocked for 50 years?

- The reference Downs and Piégay, 2019 is not listed in the bibliography list

- There are some reference in the text to sections (e.g. mention to Section 2 in page 9, or last paragraph of page 2) or headings (as mention to 5.1 in page 11). However the text is not organised in those sections and headings.

- Page 11 "Such statistical empirical relationships makeit possible..." without the dot.

Author Response

#3

The manuscript describes and condenses the role and importance of geomorphology on river rehabilitation, considering that any action to recover the rivers can only achieve rehabilitation and can not be considered restoration due to the changes in the basin conditions and river functioning drivers.

It is a very good, interesting and useful manuscript, with examples of geomorphology role and application on rivers improvement.

Thank you for this positive feedback

Another question is a whole basin perspective of rehabilitation measures, which is mentioned on upscaled geomorphology section. Some of the diagnostic tools are very fieldwork and time consuming and alternative proposals based on remote sensing, LiDAR, aerial photographs tools are explained, however these, really interesting, are very difficult to be implemented in many small, narrow river networks

One of the aspects I think it should be addressed is the duration of the post intervention monitoring. This is a key issue that should be considered when doing the planification of the actions, and having and shared database of the results of rehabilitation actions.

We added this discussion point in the text with a small paragraph.

It can be accepted in the present form after the correction of some formal and minor aspects:

- Figure 1. This figure extracted from Piégay et al 2016a, it is not well seen in the version I have seen, it is deformated, labels are not complete and the different levels of explanations are not in their correct site.

Done

- Figure 3a and 3b, why are they in the same figure? In my opinion it is more clear if they are separated in two different figures, Figure 3 and Figure 4, respectively.

Done

- Figure 6. Indicate flow direction and scale on (a). In (b) indicate the units in Downstream distance, also 10 and 5 years surface hide the previous ones, could be represent in a transparent way? In b and c, include the codes BCh, DC, TR when correspond. (c) the blue line means that the injected sediment on the injection point is completely wash out in just a year, from 6500 m3 to 0 m3? However, in the by-passed channel more than 5500 m3 are stocked for 50 years?

Done. Indeed, Vazquez et al. (2023) showed that the sediment will remain stocked in the by-passed channel, and almost no sediment will arrive at the downstream non diverted-channel after 50 years. We added this information. What the figure 7C indicates is that during the first year the sediment will be evacuated from the exact injection point, but it will propagate slowly through the bypassed channel.

- The reference Downs and Piégay, 2019 is not listed in the bibliography list

Done

- There are some reference in the text to sections (e.g. mention to Section 2 in page 9, or last paragraph of page 2) or headings (as mention to 5.1 in page 11). However the text is not organised in those sections and headings.

We reintroduced the numbering of sections and headings

- Page 11 "Such statistical empirical relationships

Done

Reviewer 2 Report

Thank you for the paper entitled "Why consider geomorphology in river rehabilitation?", submitted to the Land journal. The subject addressed is within the scope of the journal and the special issue. The outcomes could contribute to systematize our knowledge about practical and theoretical applications of fluvial geomorphology in river rehabilitation and restoration. This interesting review paper is well written as well as properly structured and creates a logical, exhaustive deduction. However, before publication in Land journal I recommend some minor revisions. My remarks are listed below.

             In the second paragraph of the introduction section please provide some specific examples of ecosystem services, which could be improved by river rehabilitation – important both for river biota and human activity (additional references should be useful). This allow to definitely improve this section.

             In the section related to TIR measurements please cite some additional papers, related to aerial thermal imagery (e.g. Torgersen et al., 2001, Airborne thermal remote sensing for water temperature assessment in rivers and streams…)

             Due to review nature of the paper in question it seems reasonable to provide some more examples of new sources of remote data, which could be useful for pre- and post-project assessment. There are some recent papers in this field (e.g. for Sentinel-1 data, Kryniecka et al., 2022, Sentinel-1 Satellite Radar Images: A New Source of Information for Study of River Channel Dynamics on the Lower Vistula River, Poland…)

             The quality of the figures (especially figure 2 and figure 3b) should be improved.

             Font size should be standardized, while citations should be made in the style recommended by the publisher (however, this can be done at the post-processing stage).

Author Response

#1

Thank you for the paper entitled "Why consider geomorphology in river rehabilitation?", submitted to the Land journal. The subject addressed is within the scope of the journal and the special issue. The outcomes could contribute to systematize our knowledge about practical and theoretical applications of fluvial geomorphology in river rehabilitation and restoration. This interesting review paper is well written as well as properly structured and creates a logical, exhaustive deduction.

Thank you for this positive feedback

However, before publication in Land journal I recommend some minor revisions. My remarks are listed below.

  • In the second paragraph of the introduction section please provide some specific examples of ecosystem services, which could be improved by river rehabilitation – important both for river biota and human activity (additional references should be useful). This allow to definitely improve this section.

We expended the paragraph as below: "(…) e.g. nutrient retention, water quality improvement, (Kaiser et al. 2020), protection of infrastructures (e.g. Brousse et al., 2021), spawning sites for high-value recreational fish species (Zeug et al., 2013), flood mitigation (Barbedo et al., 2014).”

  • In the section related to TIR measurements please cite some additional papers, related to aerial thermal imagery (e.g. Torgersen et al., 2001, Airborne thermal remote sensing for water temperature assessment in rivers and streams…)

Ok done. See below:

Airborne TIR now has long history of use to study rivers (Dugdale2016) as it allows to assess surface temperature with high precision and high resolution over large spatial scales (Torgersen et al., 2001). It has been successfully used to characterize the thermal diversity of a range of gravel-bed rivers (Wawrzyniak et al., 2013) but rarely used within a post-rehabilitation monitoring strategy.

  • Due to review nature of the paper in question it seems reasonable to provide some more examples of new sources of remote data, which could be useful for pre- and post-project assessment. There are some recent papers in this field (e.g. for Sentinel-1 data, Kryniecka et al., 2022, Sentinel-1 Satellite Radar Images: A New Source of Information for Study of River Channel Dynamics on the Lower Vistula River, Poland…)

Ok we added a full paragraph (5.3. A new era…), a figure (fig. 8) and a new author to cover this issue and did additional work to be able to illustrate explicitly how Sentinel data can help restoration monitoring.

  • The quality of the figures (especially figure 2 and figure 3b) should be improved.

Yes we did it. Figure 3a as well.

  • Font size should be standardized, while citations should be made in the style recommended by the publisher (however, this can be done at the post-processing stage).

Yes done.

Reviewer 3 Report

This paper illustrate how morphological approaches can contribute to river rehabilitation success. Morphological understanding can feed the debate at a theoretical level when considering the definition of actions, at a local project level during the diagnosis, risk assessment, and success evaluation steps, and at an up-scaled level to provide the information required to diagnose river health conditions and assess potential responses to river rehabilitation and subsequent failure or success. Different issues are considered according to the rehabilitation phase.

Questions and suggestion

Geomorphological features including morphology, genesis, material, process and age, these characteristics can affect the natural evolution process of river landforms, and the transformation and restoration process of man-made rivers. This paper main discuss why consider geomorphology in river rehabilitation. Thus, which geomorphological features can be considered in river rehabilitation? How these characteristics affect river rehabilitation, how to quantify evaluation, what way to use river rehabilitation? and so on, it is suggested to sort out these problems.

Author Response

#2

This paper illustrate how morphological approaches can contribute to river rehabilitation success. Morphological understanding can feed the debate at a theoretical level when considering the definition of actions, at a local project level during the diagnosis, risk assessment, and success evaluation steps, and at an up-scaled level to provide the information required to diagnose river health conditions and assess potential responses to river rehabilitation and subsequent failure or success. Different issues are considered according to the rehabilitation phase.

Questions and suggestion

  • Geomorphological features include morphology, genesis, material, process and age, these characteristics can affect the natural evolution process of river landforms, and the transformation and restoration process of man-made rivers.

See our reply n°3.1.

  • This paper main discuss why consider geomorphology in river rehabilitation. Thus, which geomorphological features can be considered in river rehabilitation?

Unsure what this question means. Basically, all rivers and associated geomorphological features (typically riffles, pools, bars; added in the MS) can be considered for rehabilitation given the definition of rehabilitation that we provide: we do not aim at recovering a pre-disturbance state but at improving the state of a river, by working on fluvial forms, processes or both, with the objective of supporting biodiversity, ecosystem services, river resilience, etc.

Having said that, a careful diagnosis at the river/basin scale is necessary (examples of how to conduct this diagnosis are provided in the MS) to help target most effective actions and river reaches where efforts in terms of human resources and operation costs should be put in first.

In addition, if your question refers to the spatial scale at which it can be implemented, some elements are presented in the MS as well: actions can be undertaken at different scales depending on (1) the main controlling factors that are missing/altered, (2) the degree of alteration and trajectory of the system (and thus the expectable outcomes), (3) the outcomes from the pre-rehabilitation diagnosis, (4) the balance with social acceptance, (5) the restoration objectives, etc.

  • How these characteristics affect river rehabilitation [3.1], how to quantify evaluation [3.2], what way to use river rehabilitation? [3.3] and so on, it is suggested to sort out these problems.

3.1. "How these characteristics affect river rehabilitation?" All these characteristics listed (see comment n°1 of the referee) represent more or less what the diagnosis phase aims to assess, as these shape the evolutional trajectory of a system, define its resilience and resistance to change as well as where the should cursor along be placed the rehabilitation gradient.

3.2. " how to quantify evaluation?"

Some examples are already provided, e.g. the comparative approach, the causal diagnosis as a baseline, best practice should be before-after, but we also gave an example of a control-impact strategy when no before-after strategy can be implemented.

3.3. "what way to use river rehabilitation?"

We covered the topic in pages 4 and 5 of the MS, in the sense that we showed how rehabilitation can be used in a passive vs. active way (e.g. leaving room for the river vs. removing levees, promoting self-rehabilitation vs. actively reshaping features, etc.), it can also be done in a preventive vs. curative way (e.g. protection upstream basin vs. intervening in the channel), it can be done in a trial-and-error vs. a process-based understanding fashion, etc.